# Comparison of the size of bilateral testis in children with unilateral non-communicating hydrocele and its correlation with age

Peiqiang Li👤*, Fuyun Liu, Yan Huang

Department of Pediatric Urology, The Third Affiliated Hospital of Zhengzhou University, Zhengzhou, Henan, People's Republic of China

* lipeiqiang666@zzu.edu.cn

## Abstract

### Background

Opinions on the optimal age for surgical management of hydroceles in young boys are not uniform. Scrotal ultrasonography can be used to diagnose hydroceles and measure testicular size. A comparison of bilateral testicular size with hydrocele and the change in trend with age has not been reported. We therefore aimed to analyze the bilateral testicular size of children with unilateral non-communicating hydroceles and examine the correlation between age and testicular volume.

### Methods

Non-communicating hydrocele cases in children were included. Ultrasound results, age, and diagnose time were retrospectively recorded. The bilateral testicular size was compared, and the correlation between age and testicular volume was analyzed.

### Results

There were 138 cases of non-communicating hydrocele, ranging in age from 11 to 72 months. The diagnose time ranged from 3 days to 54 months. The volume of the testis on the side of the hydrocele was larger than that on the normal side (P < 0.001). Testicular length was not different bilaterally. Testicular width and height were greater on the hydrocele side than on the normal testicular side (P<0.001). Age was positively correlated with testicular volume on the normal side (P = 0.004) but not on the hydrocele side.

### Conclusions

An important finding was that when the contralateral normal testicular volume increases with age, the testicular volume does not increase on the hydrocele side. This finding confirms the adverse effects of hydrocele on testicular growth and provides a basis for early treatment.

**Data Availability Statement:** All relevant data are within the paper and its Supporting Information files.

**Funding:** The writing of this article was supported by the Medical Science and Technology Research

Project of Henan Province provided by the Health Commission of Henan Province (Number LHGJ20190383). The funders had no role in study design, data collection and analysis, decision to publish, or preparation of the manuscript.

**Competing interests:** The authors have declared that no competing interests exist.

## Introduction

Hydrocele is a distinct fluid accumulation between the parietal and visceral layers of the tunica vaginalis [1, 2]. Primary hydroceles include communicating, non-communicating, and congenital types [1]. Congenital hydrocele in infants has a high probability of self-healing and can be treated conservatively via observation [3, 4]. Communicating hydroceles are generally treated as inguinal hernias [4]. Opinions on the optimal age for surgical management of hydroceles in young boys are not uniform, and further research is required [5]. Scrotal ultrasonography can be used to diagnose hydroceles and cryptorchidism [6]. In unilateral cryptorchidism, studies have used the opposite normal testis as a reference to evaluate the testis on the diseased side [7–9]. A similar study can be conducted using the unilateral hydrocele. A comparison of bilateral testicular volume with hydrocele and the change trend with age has yet to be reported. In this study, the bilateral testicular size of children with unilateral non-communicating hydroceles was compared, and the correlation between age and testicular volume was analyzed to explore the influence of hydroceles on testicular size.

## Materials and methods

This study was conducted at our hospital in July 2022. The clinical data of hydrocele cases from January 2014 to May 2022 were retrospectively collected. Ethical approval was obtained from the Ethics Committee of the Third Affiliated Hospital of Zhengzhou University (Medical Ethics Review No. 2022-131-01). All experimental protocols involving human data adhered to the basic principles of the Declaration of Helsinki. As it was a retrospective study, informed consent was not required.

Cases of non-communicating hydrocele in children were selected. Patients with cryptorchidism, inguinal hernia, bilateral onset, testicular tumor, communicating hydrocele, abdominoscrotal hydrocele, systemic diseases, and premature infants were excluded. Authors had access to information that could identify individual participants during data collection to select patients.

Ultrasound results, age, and diagnose time were retrospectively recorded. All testes were measured by the same sonographer using the same ultrasound instrument (GE Logic E9, General Electric Healthcare, Wauwatosa, WI, USA). ML6-15 probe was used and its frequency was 6–15 MHz. Ultrasonography was performed in the supine position in a warm, relaxed environment. The baby's cooperation was obtained through parental pacification, and no sedatives were used.

Diagnostic criteria for non-communicating hydrocele: Fluid contained within a segment of patent processus vaginalis or within the tunica vaginalis surrounding the testis, or both, no communication with the abdominal cavity, and no change in size under different body positions.

The testicular volume was calculated using the ellipsoid volume calculation formula: ($\pi/6 \times$ length $\times$ width $\times$ height) [10].

Diagnose time: The time between a caregiver finding a scrotal mass or a doctor finding a scrotal mass during a routine physical examination of a child and ultrasound examination.

### Statistical analysis

The Kolmogorov-Smirnov test was used to test for normality. Baseline characteristics were presented as medians with interquartile ranges (IQR). The testicular sizes of the hydrocele side and the normal side of the same patient were matched and compared using the paired sample rank-sum test. Spearman's rank correlation analysis was used to determine the relationship, and the LOESS method was used to draft the fitting curve. SPSS Statistics version 26 (IBM

Corp., Armonk, NY, USA) was used for statistical analyses. Statistical significance was set at two-tailed $P < 0.05$.

## Results

There were 138 cases of non-communicating hydrocele, ranging in age from 11 to 72 months, with a median (interquartile range) of 35 (24–48) months. The diagnose time ranged from 3 days to 54 months, with a median (interquartile range) of 3 (0.5–12) months. Through Kolmogorov-Smirnov test, it was found that testicular volume, length, width, height, age, and diagnose time were not normally distributed.

The testicular volume on the side of the hydrocele was larger than that on the normal side ($P < 0.001$). Testicular length was not different bilaterally. Testicular width and height were greater on the hydrocele side than on the normal testicular side ($P<0.001$) (Table 1).

Age was positively correlated with testicular volume on the normal side (r = 0.241, P = 0.004) but not on the hydrocele side (r = 0.093, P = 0.278). The results are shown in Figs 1 and 2, Table 2, respectively. There was no correlation between diagnose time and testicular volume on the normal and hydrocele sides (Table 2).

## Discussion

In the absence of disease, there is no difference in the volume between the left and right testicle [11]. Using the contralateral normal testis as a reference, we found that the volume of the testis increased with non-communicating hydrocele. Previous studies have found similar results [12, 13]. It was reported that there was no difference in testicular volume between the hydrocele group and the normal group [14]; however, this was not a comparison of the left and right sides of the same patient.

This study found no difference in length, but an increase in width and height on the hydrocele side was observed compared to that on the normal side. Hydroceles tend to round rather than flatten the ipsilateral testis [15].

Some studies have investigated the causes of increased testicular volume during hydrocele. A considerable proportion of hydrocele pressure is reportedly higher than the intraperitoneal pressure in children [16]. It has been suggested that the accretion of volume is due to increased venous and lymphatic flow impedance [13]. Hydrocele can alter the shape of the testicle through compression, increasing the pulsatility and resistive indices and causing testicular compartment syndrome [17]. As pressure in the testicle increases, the way in which it increases in volume without altering the surface area is to become more spherical.

The important finding of this study was that when contralateral normal testicular volume increases with age, testicular volume does not increase on the hydrocele side. This is the unique contribution of this study to the literature. Testicular volume is related to reproductive function [18, 19]. The pathological increase in testicular volume on the hydrocele side masks the absence of true testicular growth with age.

**Table 1. Comparison of the testes' parameters on the hydrocele and normal sides in 138 patients.**

| Factor | Hydrocele side | Normal side | P |
|---|---|---|---|
| Volume of the testis (mL), median (IQR) | 0.44 (0.35–0.58) | 0.35 (0.30–0.45) | <0.001 |
| Length of the testis (mm), median (IQR) | 15 (14–16) | 15 (14–16) | 0.283 |
| Width of the testis (mm), median (IQR) | 8 (7–9) | 7 (7–8) | <0.001 |
| Height of the testis (mm), median (IQR) | 7 (7–8) | 6 (6–7) | <0.001 |

IQR, interquartile range

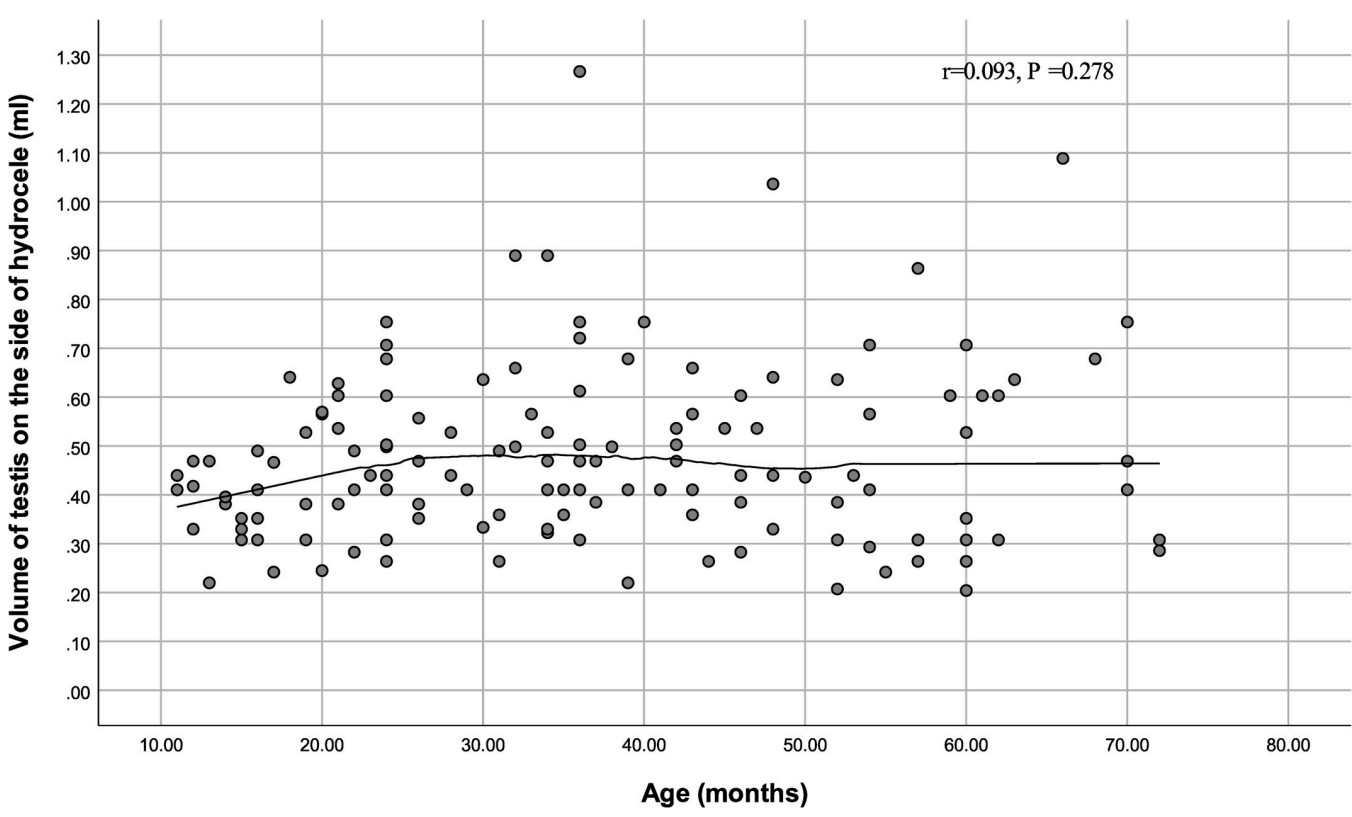

**Fig 1. Scatter plot of age and volume of testis on the hydrocele side.**

There was no correlation between diagnose time and testicular volume on the hydrocele side. We speculate that due to the limitation of the testicular white membrane, after testicular compartment syndrome [17] occurs, the testicular volume reaches a certain limit. Another possible reason is that the diagnose time is not the actual time of illness because most of which are found on physical examination.

Non-communicating hydrocele damages testicular tissue [14, 17]. The disease can be treated surgically. Studies have confirmed large testicular volumes on the hydrocele side, and a decrease in enlarged testicular volume on the hydrocele side after surgical treatment [12, 13]. Our finding provides a basis for early surgical treatment.

This study had limitations. It was a single-center, retrospective study. The age range was 11–72 months. Future prospective studies with larger sample sizes and larger age ranges will yield improved results. Further studies are needed to determine whether testicular volume restores the increasing trend with age after hydrocele treatment. Similar principles can be used to design animal experiments for further research.

## Conclusion

The testicular volume was larger on the hydrocele side than that on the normal side. Hydrocele tends to round the testis. An important finding is that when the contralateral normal testicular volume increases with age, the testicular volume does not increase on the hydrocele side. This finding confirms the adverse effects of hydrocele on testicular growth and provides a basis for early treatment.

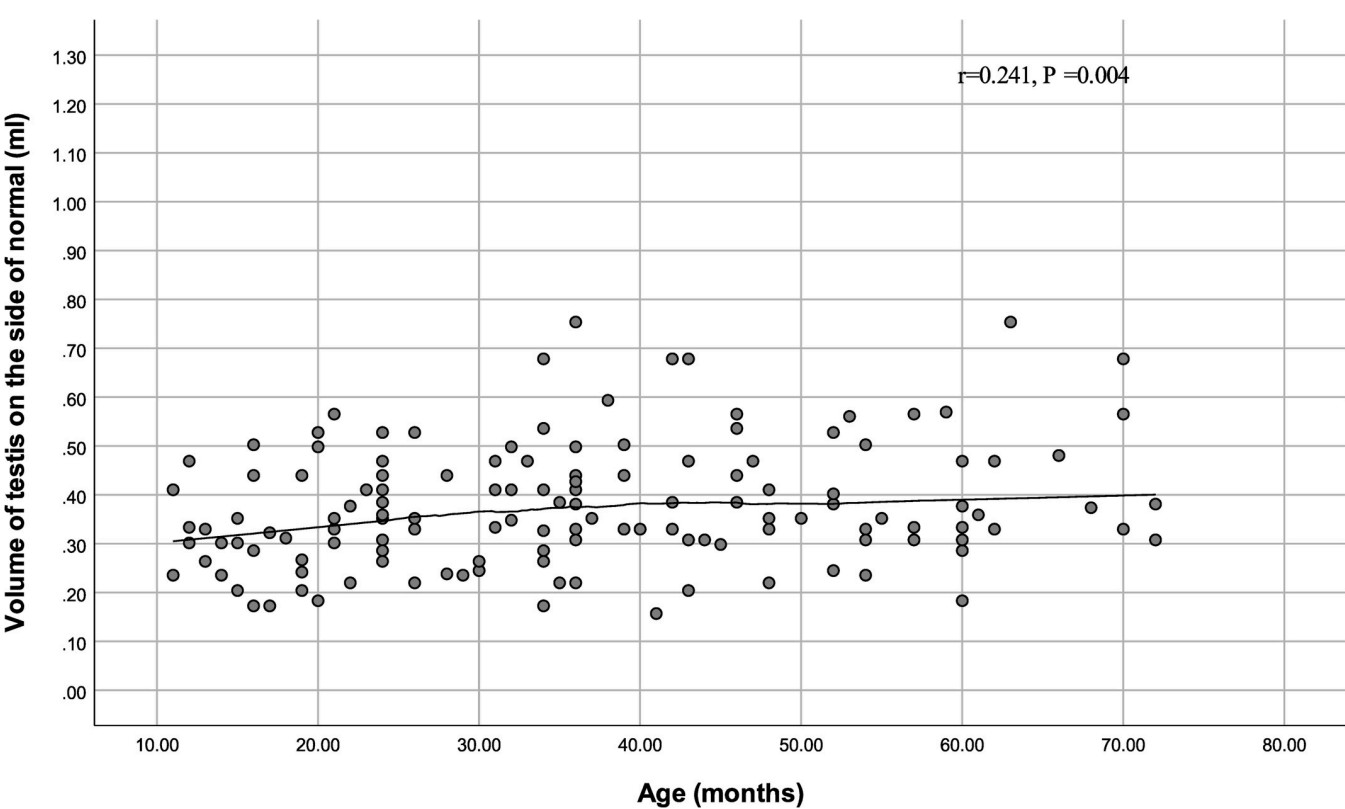

**Fig 2. Scatter plot of age and volume of testis on the normal side.**

**Table 2. Spearman's rank correlation analysis with volume of the testis in 138 patients.**

| Factor | Hydrocele side | | Normal side | |
|---|---|---|---|---|
| | r | P | r | P |
| Age | 0.093 | 0.278 | 0.241 | 0.004 |
| Diagnose time | -0.081 | 0.348 | -0.078 | 0.364 |

r, Spearman's rank correlation coefficient

## Supporting information

**S1 Data. Data of patient's age, diagnose time and ultrasound results.**
(XLSX)

## Author Contributions

**Conceptualization:** Peiqiang Li, Fuyun Liu, Yan Huang.

**Data curation:** Peiqiang Li, Fuyun Liu, Yan Huang.

**Formal analysis:** Peiqiang Li, Fuyun Liu, Yan Huang.

**Funding acquisition:** Peiqiang Li.

**Investigation:** Peiqiang Li, Fuyun Liu, Yan Huang.

**Methodology:** Peiqiang Li, Fuyun Liu, Yan Huang.

**Project administration:** Peiqiang Li, Fuyun Liu, Yan Huang.

**Resources:** Peiqiang Li, Fuyun Liu, Yan Huang.

**Software:** Peiqiang Li.

**Supervision:** Peiqiang Li.

**Validation:** Peiqiang Li, Fuyun Liu, Yan Huang.

**Visualization:** Peiqiang Li, Yan Huang.

**Writing – original draft:** Peiqiang Li, Fuyun Liu, Yan Huang.

**Writing – review & editing:** Peiqiang Li, Fuyun Liu, Yan Huang.

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
