## [Decision Letter · Decision Letter 0]

15 Nov 2022

PONE-D-22-21472Comparison of the size of bilateral testis in children with unilateral non-communicating hydrocele and its correlation with agePLOS ONE

Dear Dr. Li,

Thank you for submitting your manuscript to PLOS ONE. After careful consideration, we feel that it has merit but does not fully meet PLOS ONE’s publication criteria as it currently stands. Therefore, we invite you to submit a revised version of the manuscript that addresses the points raised during the review process.

We look forward to receiving your revised manuscript.

Kind regards,

Tai-Heng Chen, M.D.

Academic Editor

PLOS ONE

Journal Requirements: 

"The writing of this article was supported by the Medical Science and Technology Research Project of Henan Province provided by the Health Commission of Henan Province (Number LHGJ20190383). Funders had no role in the design of the study, the collection, analysis, and interpretation of data."

"Ethical approval was obtained from the Ethics Committee of the Third Affiliated Hospital of Zhengzhou University (Medical Ethics Review No. 2022-131-01). All experimental protocols involving human data adhered to the basic principles of the Declaration of Helsinki. As it was a retrospective study, informed consent was not required."

3. We note that you have stated that you will provide repository information for your data at acceptance. Should your manuscript be accepted for publication, we will hold it until you provide the relevant accession numbers or DOIs necessary to access your data. If you wish to make changes to your Data Availability statement, please describe these changes in your cover letter and we will update your Data Availability statement to reflect the information you provide

Reviewers' comments:

Reviewer's Responses to Questions

**Comments to the Author**

1. Is the manuscript technically sound, and do the data support the conclusions?

Reviewer #1: Partly

2. Has the statistical analysis been performed appropriately and rigorously? 

Reviewer #1: No

3. Have the authors made all data underlying the findings in their manuscript fully available?

Reviewer #1: Yes

4. Is the manuscript presented in an intelligible fashion and written in standard English?

Reviewer #1: No

5. Review Comments to the Author

Reviewer #1: 1. Materials and methods: The US examinations were performed by sonographers or radiologists? Whic prob was used, what was its frequency (mHz)?

2. During the sonography how the cooperation of the babies were acquired, was any sedation used?

3. Statistical analysis paragraph is so confusing and sloopy. The results of the Kolmogorov Smirnov test should be given in the results not in this section. What is LOESS method, as far as i know not a standart method in the SPSS, please give details.

4. Instead of disease time, time of the diagnosis or diagnose time would be better

5. For detecting the correlation between age and testicular volume, how long the patients were followed, this information must be added.

6. Discussion: It is a bit redundant. The authors repeated so many times that the most important finding is the cessetion of teticular growth as a result of hydrocele etc. Please summarize the discussion and conclusion sections.

6. PLOS authors have the option to publish the peer review history of their article (what does this mean?). If published, this will include your full peer review and any attached files.

Reviewer #1: **Yes: **Sonay Aydin

---

## [Author Response · Author response to Decision Letter 0]

27 Nov 2022

Journal Requirements: 

Response: Thanks again for the help of the editor for this article. We have improved the article according to PLOS ONE's style requirements.

2. Thank you for stating the following in the Acknowledgments Section of your manuscript: We note that you have provided funding information that is not currently declared in your Funding Statement. However, funding information should not appear in the Acknowledgments section or other areas of your manuscript. We will only publish funding information present in the Funding Statement section of the online submission form. 

Please remove any funding-related text from the manuscript and let us know how you would like to update your Funding Statement. Currently, your Funding Statement reads as follows: "Ethical approval was obtained from the Ethics Committee of the Third Affiliated Hospital of Zhengzhou University (Medical Ethics Review No. 2022-131-01). All experimental protocols involving human data adhered to the basic principles of the Declaration of Helsinki. As it was a retrospective study, informed consent was not required."Please include your amended statements within your cover letter; we will change the online submission form on your behalf.

Response: I apologize for the trouble caused by my mistake. I removed funding-related text from the manuscript and amended statements within my cover letter. Please help me change the online submission form.

3. We note that you have stated that you will provide repository information for your data at acceptance. Should your manuscript be accepted for publication, we will hold it until you provide the relevant accession numbers or DOIs necessary to access your data. If you wish to make changes to your Data Availability statement, please describe these changes in your cover letter and we will update your Data Availability statement to reflect the information you provide

Response: I apologize for the trouble caused by my mistake. I described the change in my cover letter. Since the data is not uploaded to the public repository, Data Availability statement needs to be amended. Data Availability statement: All relevant data are within the paper and its Supporting information files.

4. In your Data Availability statement, you have not specified where the minimal data set underlying the results described in your manuscript can be found. "Upon re-submitting your revised manuscript, please upload your study’s minimal underlying data set as either Supporting Information files or to a stable, public repository and include the relevant URLs, DOIs, or accession numbers within your revised cover letter. 

Response: I apologize again for the trouble caused by my mistake. I described the change in my cover letter. Since the data is not uploaded to the public repository, Data Availability statement needs to be amended. Data Availability statement: All relevant data are within the paper and its Supporting information files. Please help me update my Data Availability statement.

Reviewers' comments:

Reviewer's Responses to Questions

Comments to the Author

1. Is the manuscript technically sound, and do the data support the conclusions?

Reviewer #1: Partly

Response: Once again, we thank the reviewer for this insightful comment on the article. We revised the article to make it more technically sound, and the data supports the conclusion.

2. Has the statistical analysis been performed appropriately and rigorously?

Reviewer #1: No

Response: I'm sorry for the confusion in statistics. As for this problem, it has been improved according to the description of "5. Review Comments to the AuthoReviewer # 1: 3. Statistical analysis paragraph is… ". 

The description of statistics is further improved (Lines 96-97 of the manuscript). The results of Kolmogorov Smirnov test have been transferred to the results section (Lines 109-110 of the manuscript). In order to display the trend of data change more intuitively, we have made a LOESS curve on the scatter plot. This function is rarely used. We have searched for it for a long time and found it is one of the functions of SPSS software. The official description is as follows: https://www.ibm.com/support/pages/node/473567 . The drawing process of LOESS curve is shown in the form of pictures at the end of this WORD document.

3. Have the authors made all data underlying the findings in their manuscript fully available?

Reviewer #1: Yes

Response: All relevant data are within the Supporting information file. In order to make the data easier to be opened by the public, the Supporting data is uploaded in xlsx format with reference to the situation similar to PLoS ONE.

4. Is the manuscript presented in an intelligible fashion and written in standard English?

Reviewer #1: No

Response: We apologize for this deficiency. Our manuscript has been professionally edited by a language editing service (editage, Cactus Communications Services Pte. Ltd). The discussion and conclusion are simplified and improved.

5. Review Comments to the Author

Reviewer #1: 1. Materials and methods: The US examinations were performed by sonographers or radiologists? Whic prob was used, what was its frequency (mHz)?

Response: Thank you for pointing out what needs to be improved. Added relevant information (Lines 79-82 of the manuscript). “All testes were measured by the same sonographer using the same ultrasound instrument (GE Logic E9, General Electric Healthcare, Wauwatosa, WI, USA). ML6-15 probe was used and its frequency was 6-15 MHz.”

2. During the sonography how the cooperation of the babies were acquired, was any sedation used?

Response: Added relevant information (Lines 83-84 of the manuscript). “The baby's cooperation was obtained through parental pacification, and no sedatives were used.”

3. Statistical analysis paragraph is so confusing and sloopy. The results of the Kolmogorov Smirnov test should be given in the results not in this section. What is LOESS method, as far as i know not a standart method in the SPSS, please give details.

Response: I'm sorry for the confusion in statistics. The description of statistics is further improved (Lines 96-97 of the manuscript). The results of Kolmogorov Smirnov test have been transferred to the results section (Lines 109-110 of the manuscript). In order to display the trend of data change more intuitively, we have made a LOESS curve on the scatter plot. This method is rarely used. We have searched for it for a long time and found it is one of the functions of SPSS software. The official description is as follows: https://www.ibm.com/support/pages/node/473567 . The drawing process of LOESS curve is shown in the form of pictures at the end of this WORD document.

4. Instead of disease time, time of the diagnosis or diagnose time would be better

Response: This is a very good suggestion. We have replaced “disease time” with “diagnose time” in the article.

5. For detecting the correlation between age and testicular volume, how long the patients were followed, this information must be added.

Response: Thank you for your valuable research direction. Because this is a retrospective study, we regret that there is no follow-up data in these case data, so we can only use these existing data to form this article. Further prospective follow-up study can be conducted according to your prompts. In the last paragraph of the discussion, a forward-looking statement was made (Lines 165-169 of the manuscript).

6. Discussion: It is a bit redundant. The authors repeated so many times that the most important finding is the cessetion of teticular growth as a result of hydrocele etc. Please summarize the discussion and conclusion sections.

Response: Thank you for pointing out this shortcoming. I summarized the discussion and conclusion and the unnecessary duplication of important findings of this study was simplified (Lines 44,47,141,150,154,159-160,164,175 of the manuscript). More than 20% of unnecessary words in this part have been deleted without affecting the expression of meaning.

Supplement to 5. Review Comments to the Author 3.

The drawing process of LOESS curve is shown in the form of pictures.

---

## [Decision Letter · Decision Letter 1]

20 Dec 2022

Comparison of the size of bilateral testis in children with unilateral non-communicating hydrocele and its correlation with age

PONE-D-22-21472R1

Dear Dr. Li,

We’re pleased to inform you that your manuscript has been judged scientifically suitable for publication and will be formally accepted for publication once it meets all outstanding technical requirements.

Kind regards,

Tai-Heng Chen, M.D.

Academic Editor

PLOS ONE

Reviewers' comments:

Reviewer's Responses to Questions

**Comments to the Author**

1. If the authors have adequately addressed your comments raised in a previous round of review and you feel that this manuscript is now acceptable for publication, you may indicate that here to bypass the “Comments to the Author” section, enter your conflict of interest statement in the “Confidential to Editor” section, and submit your "Accept" recommendation.

Reviewer #1: All comments have been addressed

2. Is the manuscript technically sound, and do the data support the conclusions?

Reviewer #1: Yes

3. Has the statistical analysis been performed appropriately and rigorously? 

Reviewer #1: Yes

4. Have the authors made all data underlying the findings in their manuscript fully available?

Reviewer #1: Yes

5. Is the manuscript presented in an intelligible fashion and written in standard English?

Reviewer #1: Yes

6. Review Comments to the Author

Reviewer #1: The revisions are satisfying thank you. All of the concerns have been adressed, the latest form seems to be acceptable.

7. PLOS authors have the option to publish the peer review history of their article (what does this mean?). If published, this will include your full peer review and any attached files.

Reviewer #1: **Yes: **SONAY AYDIN

---

## [Editor Report · Acceptance letter]

23 Dec 2022

PONE-D-22-21472R1 

Comparison of the size of bilateral testis in children with unilateral non-communicating hydrocele and its correlation with age 

Dear Dr. Li:

I'm pleased to inform you that your manuscript has been deemed suitable for publication in PLOS ONE. Congratulations! Your manuscript is now with our production department. 

Kind regards, 

on behalf of

Dr. Tai-Heng Chen 

Academic Editor

PLOS ONE